# An Isoform of the Eukaryotic Translation Elongation Factor 1A (eEF1a) Acts as a Pro-Viral Factor Required for Tomato Spotted Wilt Virus Disease in *Nicotiana benthamiana*

**DOI:** 10.3390/v13112190

**Published:** 2021-10-30

**Authors:** Tieme A. Helderman, Laurens Deurhof, André Bertran, Sjef Boeren, Like Fokkens, Richard Kormelink, Matthieu H. A. J. Joosten, Marcel Prins, Harrold A. van den Burg

**Affiliations:** 1Molecular Plant Pathology, Swammerdam Institute for Life Sciences (SILS), University of Amsterdam, Science Park 904, 1098 XH Amsterdam, The Netherlands; t.a.helderman@uva.nl (T.A.H.); l.fokkens@uva.nl (L.F.); Marcel.prins@keygene.com (M.P.); 2Laboratory of Phytopathology, Department of Plant Sciences, Wageningen University and Research, Droevendaalsesteeg 1, 6708 PB Wageningen, The Netherlands; Laurens.deurhof@wur.nl (L.D.); Matthieu.joosten@wur.nl (M.H.A.J.J.); 3Laboratory of Virology, Department of Plant Sciences, Wageningen University and Research, Droevendaalsesteeg 1, 6708 PB Wageningen, The Netherlands; Andre.machadobertran@wur.nl (A.B.); Richard.kormelink@wur.nl (R.K.); 4Laboratory of Biochemistry, Department of Agrotechnology and Food Sciences, Wageningen University and Research, Stippeneng 4, 6708 WE Wageningen, The Netherlands; sjef.boeren@wur.nl; 5KeyGene N.V., Agro Business Park 90, 6708 PW Wageningen, The Netherlands

**Keywords:** eukaryotic translation elongation factor 1A (eEF1A), host factor, nanoLC-MS/MS, SDS-PAGE-based proteomics, pro-viral factor, ribonucleoproteins (RNP), susceptibility factor, *Tomato spotted wilt orthotospovirus* (TSWV)

## Abstract

The tripartite genome of the negative-stranded RNA virus *Tomato spotted wilt orthotospovirus* (TSWV) is assembled, together with two viral proteins, the nucleocapsid protein and the RNA-dependent RNA polymerase, into infectious ribonucleoprotein complexes (RNPs). These two viral proteins are, together, essential for viral replication and transcription, yet our knowledge on the host factors supporting these two processes remains limited. To fill this knowledge gap, the protein composition of viral RNPs collected from TSWV-infected *Nicotiana benthamiana* plants, and of those collected from a reconstituted TSWV replicon system in the yeast *Saccharomyces cerevisiae*, was analysed. RNPs obtained from infected plant material were enriched for plant proteins implicated in (*i*) sugar and phosphate transport and (*ii*) responses to cellular stress. In contrast, the yeast-derived viral RNPs primarily contained proteins implicated in RNA processing and ribosome biogenesis. The latter suggests that, in yeast, the translational machinery is recruited to these viral RNPs. To examine whether one of these cellular proteins is important for a TSWV infection, the corresponding *N. benthamiana* genes were targeted for virus-induced gene silencing, and these plants were subsequently challenged with TSWV. This approach revealed four host factors that are important for systemic spread of TSWV and disease symptom development.

## 1. Introduction

Viruses belonging to the family *Tospoviridae* cause severe problems in agriculture worldwide. *Tomato spotted wilt orthotospovirus* (TSWV) is the type-species of this plant-infecting virus family. TSWV can replicate in a wide range of plant species, including monocotyledons and dicotyledons [1,2]. The virus is transmitted in a propagative manner by several thrips species (order *Thysanoptera*) with the main vector, *Frankliniella occidentalis*, now having spread globally. As with all viruses, TSWV needs to manipulate cellular processes by hijacking host proteins for its own needs, such as replication and cell-to-cell spread [3,4]. Consequently, genetic variation in any of these host factors could turn a natural host species into a non-host when the genetic variation hampers the viral infection cycle. These genetic variants are often recessive alleles, and they are now referred to as ‘susceptibility (S) genes’. In fact, engineered mutant alleles of these S genes are now widely regarded to be an effective means to obtain sustainable resistance to plant viruses [5,6]. To this date, only four candidate S genes have been reported for TSWV, that is, *9-lipoxygenase* (*9LOX*) and *α-dioxygenases* (*α-DOX*) in Arabidopsis [7], *root hair defective 3* (*rhd3*) also in Arabidopsis [8], and *suppressor of the G2 allele of skp1* (*SGT1*) in *N. benthamiana* [9]. In all cases, downregulation or a stable genetic variant of these genes was reported to result in attenuation of TSWV disease symptoms. Here, we report on the identification of novel candidate S genes for TSWV, using a proteomics approach that aimed to identify the host protein associated with viral ribonucleoprotein complexes (RNPs) and confirmed, by gene silencing, their role as a pro-viral host factor for TSWV.

The genome of TSWV is composed of three single-stranded RNA (ssRNA) segments, named after their relative sizes (i.e., large (L), medium (M), and small (S)) [10]. The uncapped 5’- and 3’-termini of these viral RNAs are self-complementary, resulting in the formation of an RNA duplex panhandle structure that gives rise to pseudo-circularisation of the viral genome segments. These RNA segments are tightly encapsulated by nucleocapsid protein (hereafter called the N protein), and they also contain (one or more copies of) the viral RNA-dependent RNA polymerase (RdRp, also referred to as L). These viral RNPs are, by themselves, infectious when introduced into plant cells (by rub-inoculation or protoplast transfection) [11,12]. Replication of the viral genome is orchestrated by the viral RdRp that binds to highly conserved promoter elements in the panhandle structure, from where it transcribes and replicates the individual viral segments [13]. RNPs engage in a replication–transcription cycle, during which near-genomic length L mRNA and sub-genomic length mRNAs from the ambisense M- and S-RNA are produced. These viral mRNAs are translated into five viral proteins, i.e., the RdRp/L protein, the M RNA-encoded cell-to-cell movement protein (NSm) and precursor protein of the viral glycoproteins (GP), and the S-RNA-encoded RNA silencing suppressor (NSs) and the N protein. Viral genomic RNA (vRNA) and viral antigenomic/complimentary RNA (vcRNA) strands resulting from replication are encapsulated by the N and L proteins into functional RNPs [14]. These RNPs are transported in a NSm-dependent manner via the plasmodesmata to neighbouring cells [15], or they are translocated by the two glycoprotein moieties from the endoplasmic reticulum to the Golgi complex, where they form membrane-enveloped virus particles [16,17,18]. 

Replication of TSWV RNPs likely requires dimerisation of two L proteins, as was shown for *La Crosse orthobunyavirus* (LACV), which is a mammal-infecting distant relative of TSWV [19,20]. The transcription of TSWV mRNAs is initiated by a process called ‘cap-snatching’. During this process, the L protein removes 5’-capped non-viral leader sequences from capped host mRNAs and uses them to prime the synthesis of viral mRNAs [21]. The capped viral mRNA is then loaded into the ribosome to allow synthesis of viral proteins. For some (animal-infecting) bunyaviruses, the process of transcription and translation is coupled to prevent premature transcription termination [22,23], but this is not the case for TSWV, as transcription continues even when protein synthesis is inhibited by the pharmaceuticals edeine or cycloheximide [24]. Conceivably, different host factors play important roles during the transcription and translation of the viral mRNAs. So far, only the eukaryotic translation Elongation Factor 1A (eEF1A) protein family was found to facilitate TSWV RNA synthesis in extracts from tobacco (*Nicotiana tabacum*) protoplasts [25].

In the past, bakers’ yeast (*Saccharomyces cerevisiae*) has successfully been used as a (non-natural) model system to identify replication-associated factors that impact the host–virus interaction [26]. Recently, others have described a TSWV replicon system for yeast, even though yeast is a non-host for this virus [27]. Using this artificial replicon system, it was demonstrated that expression of two yeast codon-optimised viral gene constructs, i.e., the RdRp and the nucleocapsid protein (L and N protein, respectively), results in the formation of viral RNPs in the presence of a DNA construct that, upon transcription, mimics the viral complementary genomic S segment (vcRNA). Interestingly, using the vcRNA of the TSWV S segment as a template, these artificial yeast RNPs were capable of replicating this vcRNA into the vRNA. However, when the gene-coding part of this genomic S RNA was replaced by the reverse complement of the *YFP* (*Yellow fluorescent protein*) gene, RNA replication was observed as well, but transcription and subsequent translation of the *YFP* open reading frame (ORF) did not occur. Apparently, the L protein is capable of replicating viral RNAs in the yeast replicon system, but yeast lacks (host) factors to allow subsequent viral transcription, while such factors are clearly present in the insect vector and host plants to initiate transcription [21,28]. Irrespective of this incompatibility in yeast, yeast factors linked to transcription and/or translation might still be associated with the replicon-derived viral RNPs in yeast.

Here, we hypothesised that viral RNPs from TSWV-infected plants are likely to contain host proteins involved in the TSWV infection process. Using peptide mass fingerprinting, the protein composition was characterised of the two aforementioned types of viral RNPs, namely (a) RNPs isolated from TSWV-infected plants and (b) RNPs generated with the TSWV replicon system in yeast to identify conserved host factors involved in viral RNA replication [27]. Interestingly, gene ontology (GO) enrichment tests for the genes encoding the proteins associated with the yeast RNP fractions revealed that proteins involved in cellular processes connected with translation were associated with the partially functional yeast RNPs. In contrast to the yeast RNPs, the plant RNPs were enriched in host proteins predicted to act during late stages of the viral infection. Akin to a reverse genetics approach, we subsequently assessed whether knock-down of the genes encoding the co-purifying host factors, resulted in reduced TSWV infectivity. To this end, the closest homologues of these genes in *N. benthamiana* were targeted by virus-induced gene silencing (VIGS), and the silenced plants were then challenged with TSWV. This work revealed that silencing of *Charged multivesicular body protein 1* (*CHMP1*), *β-1,3-glucanase (βGLU)*, *Ammonium transporter 1 (AMT1)*, and a homologue of the *Eukaryotic translation elongation factor 1 alpha* (*eEF1A*) (NbS00023178g0001.1) suppressed both TSWV disease symptom development and systemic spread of the virus to different degrees. Furthermore, the data indicated that silencing one clade of the *eEF1A* gene family strongly suppressed TSWV accumulation and disease symptom development in *N. benthamiana*, suggesting that the gene products act as important susceptibility factors for TSWV.

## 2. Materials and Methods 

### 2.1. Viral Strains and Plant Material

Throughout this study, the Brazilian TSWV isolate BR-01 was used [29]. TSWV was periodically passed through *Emilia sonchifolia* as described [30] and was maintained in *N. benthamiana* for disease trials. TSWV inoculations were performed by homogenising infected *N. benthamiana* leaf material in inoculation buffer (10 mM Na_2_HPO_4_/NaH_2_PO_4_ pH 7.0, 10 mM Na_2_SO_3_), and the extract was rub-inoculated on *N. benthamiana* leaves after dusting the leaves with 500-mesh carborundum. Four- to five-week-old *N. benthamiana* plants were used for the TSWV disease assays. For virus-induced gene silencing (VIGS), two-week-old *N. benthamiana* plants were pre-infected with Tobacco rattle virus (TRV) expressing a *N. benthamiana* gene fragment [31]. All plants were grown on compost soil in a controlled greenhouse environment with 16 h of light (636–1060 µmol m^−2^ s^−1^) at 22 °C and 8 h darkness at 20 °C and at ~65% relative humidity.

### 2.2. RNP Purification from TSWV-Infected N. benthamiana

*N. benthamiana* leaf homogenates were prepared from TSWV-infected or control (mock-inoculated) plants with minor modifications [32,33]. Briefly, systemically infected leaves or comparable leaves from the control plants were sampled of ten *N. benthamiana* plants at 14 days post TSWV rub-inoculation. The harvested material was ground in a stainless-steel Waring blender with 4 mL of ice-cold, freshly prepared extraction buffer (TAS-E) (100 mM Tris-HCl pH 8.0; 10 mM EDTA; 0.1% *w*/*v* cysteine; and 10 mM Na_2_SO_3_) per gram leaf material. The resulting crude homogenate was filtered over two layers of cheesecloth, and the filtrate was then centrifuged at 1500× *g* at 4 °C for 10 min. The supernatant was centrifuged a second time at 31,000× *g* at 4 °C for 30 min. The obtained pellet was resuspended in 4 mL of freshly prepared resuspension buffer (TAS-R ) (10 mM Tris-HCl pH 7.9; 10 mM EDTA; 10 mM Na_2_SO_3_; 0.1% *w*/*v* cysteine; 10 mM glycine; 1% *v*/*v* Nonidet P40) per ten grams of original leaf material by gently stirring the suspension at 4 °C for one hour. The suspension was then layered on top of 30 mL of a 30% *w*/*v* sucrose cushion in TAS-R and subjected to ultra-centrifugation at 61,000× *g* (4 °C) for one hour. The RNP-containing pellet was finally resuspended in 0.5 mL 10 mM citrate buffer (pH 6.0) per 25 g of original leaf material and stored at −80 °C until further usage.

### 2.3. S. cerevisiae Expressing a TSWV Replicon System

To isolate viral RNPs from yeast, yeast vectors containing the cDNA of the complete genomic TSWV *S* segment (vcRNA) and the yeast-codon optimised cDNA clones of the TSWV viral genes *N* and *L* [27] were introduced in the yeast strain THY.AP4, as described [34]. Transformed yeast cells were plated on complete selective media (CSM) to maintain the plasmids, i.e., minimal medium (1.7 g/L yeast nitrogen base without amino acids, 5 g/L ammonium sulphate, 20 g/L glucose, and 1.5 g/L dropout mix, adjusted to pH 6.0 with KOH, including 20 g/L amino acid free agar when solid media is needed), without the amino acids Leu, Trp, and/or Ura and incubated at 30 °C. Yeast cells harbouring the plasmids were cultured in liquid CSM medium at 30 °C with 200 rpm agitation, and the liquid culture was used to start a 1 L overnight culture, grown at 30 °C with 150 rpm agitation. The viral genes were induced by adding CuSO_4_ at a final concentration of 0.2 mM to the yeast culture, and culturing was continued for an additional 8 h at 30 °C. The yeast cells were collected by centrifugation at 3000× *g* for 15 min at 4 °C, and the obtained cell pellet was snap frozen in liquid N_2_. Subsequently, the cells were pulverised with a pestle in a liquid N_2_-cooled mortar, and the lysed cells were resuspended in 10 mL of ice-cold TAS-E buffer per gram of cell pellet. The subsequent RNP purification was performed as described in Section 2.2. The isolated RNP fraction was inspected using immuno-electron microscopy as described [12].

### 2.4. In-Gel Tryptic Digestion and Mass-Spectrometry (MS) Analysis

To determine the protein concentration of the RNP fractions, a bicinchoninic acid (BCA) assay was performed with Bovine serum albumin (BSA) as protein standard. Samples were prepared in triplicate. In total, 80 µg of protein extract from each sample was individually mixed with SDS loading buffer (60 mM Tris-HCl pH 6.8, 10% *v*/*v* glycerol, 2% *w*/*v* SDS, 5% *v/v* β-mercaptoethanol, 0.01% *w*/*v* bromophenol blue). The protein samples were then denatured by heating them at 85 °C for 5 min and were loaded on a 4–20% Mini-PROTEAN® TGX™ Precast Protein gradient gel (Bio Rad, Hercules, CA, USA), followed by a short electrophoresis run at low voltage to fix the proteins in the gel. To visualise the proteins, the gel was stained with Coomassie PageBlue (ThermoFischer, Waltham, MA, USA) for one hour, according to the suppliers’ instructions, and the protein bands were excised. To reduce the disulphide bonds, the obtained gel pieces were incubated in 50 mM ammonium bicarbonate pH 8.0 (ABC buffer), supplemented with 10 mM DTT at 60 °C for one hour. Subsequently, 20 mM acrylamide in 100 mM Tris-HCl (pH 8.0) was added, and the samples were left at room temperature in the dark for 1 h. Next, the gel pieces were cut in cubes of approximately 1 mm^3^ and transferred to a protein LoBind® microcentrifuge tube (Eppendorf, Hamburg, Germany). Tryptic digestion was performed as described [35] with minor adaptations. In-gel tryptic digestion was performed by adding 5 ng/µL Trypsin (sequencing grade, Roche, Basel, Switzerland) in ABC buffer and incubating the mixture overnight with gentle rocking at 20 °C. Protein digestion was terminated by acidifying the mixture to a pH below 3 by adding 1 to 3 µL of 10% *v*/*v* trifluoroacetic acid in water. The peptide digest mixtures were then passed over home-made C18 disc-based µColumns [36], and the bound peptides were eluted in 50% acetonitrile with 0.05% formic acid in demineralised water. The peptide eluates were collected in LoBind® tubes (Eppendorf, Hamburg, Germany) and concentrated in a Speed-Vac at 45 °C and 25 mbar until the total volume was reduced to 25 µL to evaporate the acetonitrile. To all samples 0.05% formic acid in demineralised water was added to a final volume of 50 µL.

The peptide samples were analysed using a nanoLC-Q Exactive™ HF-X Hybrid Quadrupole-Orbitrap LC-MS (ThermoFisher, Waltham, MA, USA). Five µL of the peptide samples were loaded directly onto a 0.10 × 250 mm ReproSil-Pur 120 C18-AQ 1.9 µm beads analytical column (prepared in-house), at a constant pressure of 825 bar (with a flow rate of circa 700 nL/min), with 1 mL/L HCOOH in water and eluted at a flow rate of 0.5 µL/min, with a 50 min linear gradient from 9% to 34% acetonitrile in water, containing 1 mL/L formic acid, with a Thermo EASY nanoLC1000. An electrospray potential of 3.5 kV was applied directly to the eluent via a stainless steel needle fitted into the waste line of the micro cross that was connected between the nLC pump and the analytical column. A nanoBlow was installed to prevent ions entering the MS during equilibration, while loading the sample and for the first 5 min after injection [37]. Full scan positive mode FT-MS spectra were measured between *m*/*z* 380 and 1400 on a Q-Exactive HFX mass spectrometer (Thermo electron, San Jose, CA, USA) at a resolution of 60,000. MS and MS/MS automatic gain control (AGC) targets were set to 3106 and 50,000, respectively, and maximum ion injection times of 50 ms (MS) and 25 ms (MS/MS) were used. HCD fragmented (isolation width 1.2 *m*/*z*, 24% normalised collision energy) MS/MS scans of the 25 most abundant 2+ or 3+ charged peaks in the MS scan were recorded in the data-dependent mode (threshold 1.2 × 10^5^, 15 s exclusion duration for the selected *m*/*z* +/− 10 ppm).

The Andromeda search engine of the MaxQuant software [38,39] was used to identify individual peptides and to calculate the relative protein abundance based on the label-free quantification (LFQ) method for the corresponding proteins. Peptide identifications from MS/MS spectra were obtained by scanning a combined protein database composed of *N. benthamiana* proteins (http://bti.cornell.edu/nicotiana-benthamiana/, version V.0.4.4, accessed on 14 February 2017), consisting of 76,379 protein sequence entries, a list of TSWV protein entries (http://uniprot.com/, accessed on 5 December 2017) with 2108 entries, and a list of in-house contaminants (63 protein entries) [40]. For *S. cerevisiae* peptide identifications, the bakers’ yeast proteome of 2013 (UP000002311), provided by UniProt was used (consisting of 6626 protein sequence entries). The LFQ values of the individual proteins were used for data filtering using Perseus v1.6.3.4 software [41]. Protein identifications were accepted when a protein was detected by at least two tryptic peptides, of which at least one was unique and one peptide unmodified. The LFQ values of the identified proteins were log_10_ transformed prior to further statistical analyses. Missing value imputations of protein intensities were performed by replacing the missing values for the lowest number subtracted by 0.1. To be included in our subject analyses, an identified protein had to be present in at least two out of three biological replicates per sample group. Student’s t-test, controlled for a permutation-based false discovery rate (FDR) of <0.05, was applied to determine significant differences (more than a two-fold change) for the protein abundance between sample groups. The original MS/MS dataset of the RNP proteomics has been deposited in the ProteomeXchange Consortium via the PRIDE [42] partner repository, with the dataset identifier PXD026246.

### 2.5. Functional Enrichment Analyses and eEF1A Phylogeny

Because no gene ontology (GO) annotation was available for *N. benthamiana*, for each *N. benthamiana* protein identified in the MS/MS analyses, the best hit (Blastp; lowest E-value and high similarity score) in the *Arabidopsis thaliana* genome was selected. GO enrichment analyses were performed using the singular enrichment analysis (SEA) tool in AgriGO v2.0 [43], comparing the identified plant proteins to the *A. thaliana* reference proteome or the list of RNP-enriched proteins in yeast against the *S. cerevisiae* reference proteome. GO terms were accepted using hypergeometric test with a correction for multiple-testing (Yekutieli FDR under dependency) with a *Q*-value cut-off of 0.05.

For the plant RNPome, we identified orthologs in the yeast RNPome as follows: we used eggNOG mapper V2 [44] to assign RNP-associated plant and yeast proteins identified in the MS/MS analyses to eukaryotic orthologous groups (eggNOG v5.0 [45]). Plant and yeast proteins that were assigned to the same orthologous group (KOG) were considered orthologs.

We used nucleotide blast searches to identify close homologues of the *N. benthamiana eEF1A* gene (NbS00023178g0001.1) in the QUT *N. benthamiana* transcriptome database v6.1 (Brisbane, Australia). Only full-length ORFs were retained for further analysis. Similarly, close homologues of *N. benthamiana eEF1A* were obtained from different plant genomes (Arabidopsis, *Capsicum annuum*, *Nicotiana attenuata*, *Oryza sativa*, *Solanum lycopersicum*, and *S. tuberosum*) by searching the Phytozome database v12.1.6 (Berkeley, CA, USA). The obtained DNA sequences for the *eEF1A* ORFs were aligned using MUSCLE v3.8.31, and gaps and/or poorly aligned regions were removed using Gblocks v0.91b [46]. The phylogenetic gene tree was then reconstructed using a maximum-likelihood (ML) approach as implemented in RAxML v0.9.0 with an unpartitioned DNA model and the HKY substitution matrix with a proportion of invariant sites and 4 gamma-distributed rate categories [47]. The reliability of the internal branches was assessed by automatic bootstrapping method (with a bootstopping cut-off of 0.03). The graphical representation of the phylogenetic tree was generated using FigTree v1.4.4 (http://tree.bio.ed.ac.uk/software/figtree/, accessed on 18 September 2020). The Venn diagram was generated with a webtool (http://bioinformatics.psb.ugent.be/webtools/Venn/, accessed on 27 March 2020).

### 2.6. Virus-Induced Gene Silencing and Serological Detection of TSWV

For virus-induced gene silencing (VIGS), DNA fragments were designed of ±300 nucleotides in length (https://vigs.solgenomics.net/, accessed on 6 December 2018) [48] and amplified from *N. benthamiana* cDNA using the PCR primers given in Appendix A. In total, 61 fragments were successfully amplified from cDNA. These fragments were cloned into the *Sma*I restriction site of the TRV2 cloning vector pYL156TE [31], and the resulting constructs were verified by DNA sequencing and subsequently introduced into *Agrobacterium tumefaciens* strain GV3101. The different TRV2 constructs (at an OD_600_ of 1.6) were mixed in a 1:1 ratio with a suspension of an *A. tumefaciens* strain containing pTRV1 (OD_600_ = 1.6), resulting in a final OD_600_ of 1.6 for the mixture. This bacterial suspension was then syringe-infiltrated into the first two true leaves of two-week-old *N. benthamiana* plants. As a negative control for the VIGS experiment, a TRV2 construct was used that targets a non-plant gene, i.e., *β-glucuronidase* from *Escherichia coli* (TRV::*GUS*) [49]. Two weeks post TRV inoculation, the plants were rub-inoculated with sap containing TSWV virions, and TSW disease progression was monitored during the next two weeks.

TSWV viral titres were determined for all plants lacking visual TSWV disease symptoms, using a double-antibody sandwich enzyme-linked immunosorbent assay (DAS-ELISA), as described previously [50]. Briefly, 96-well flat bottom medium-binding ELISA microtiter plates (Greiner Bio One, Alphen aan den Rijn, The Netherlands) were coated with a rabbit-raised polyclonal antibody against the nucleocapsid protein (1:1000 *v*/*v*) [12] in coating buffer (50 mM sodium bicarbonate pH 9.6) overnight at 4 °C. Before adding protein extracts, the coated microtiter plates were washed twice with PBS-T (10 mM potassium phosphate buffer pH 7.0, 140 mM NaCl, 0.1% *v*/*v* Tween-20), rinsed twice with distilled water, and dried upside down on a paper towel. Using a 5 mm diameter leaf puncher, six leaf disks were taken of the youngest fully expanded leaf near the plant apex that lacked TSWV disease symptoms. Leaf disks were combined in a 2.0 mL microcentrifuge tube containing 200 µL PBS-T buffer and three 3 mm diameter steel metal balls and directly placed on ice. Samples were homogenised using a bead mill homogeniser (Mixer Mill MM 400, Retsch, Haan, Germany) at 25 Hz for 1 min in pre-cooled sample holders (−20 °C) and the homogenate was cleared by centrifugation at 12,000× *g* for 15 s at 4 °C. Subsequently, 100 µL of the supernatant was transferred to the antibody-coated plate, and the plates were incubated overnight at 4 °C without agitation. The following day, the microtiter plate was washed as described above. Subsequently, 200 µL of PBS-T, supplemented with rabbit-raised polyclonal anti-N antibody conjugated to Alkaline phosphatase (1:1000 *v*/*v*) [51], was applied to each well. Following antigen binding (two hours at 37 °C), the plates were rinsed as described before. Phosphatase substrate (Sigma-Aldrich, St. Louis, MO, USA) was dissolved to 1 mg/mL in substrate buffer (10% diethanolamine pH 9.8, 0.02% sodium azide) and 200 µL was added to each well, followed by 30 min of incubation at room temperature in the dark. The absorbance of each well was measured at 405 nm with 30 min interval for two hours using a FLUOstar Optima plate reader (BMG LabTech, Ortenberg, Germany). Significant differences were determined using Student’s t-test.

### 2.7. Gene Expression Analysis Using Real Time RT-PCR

To quantify the transcript levels of the plant genes studied, or to determine the viral RNA levels, a total of 100 mg of systemic leaf tissue was collected near the apex from 6 week old *N. benthamiana* plants at two weeks post TSWV inoculation (=four weeks after TRV inoculation). Following tissue grinding with a bead mill homogeniser, total RNA was extracted using TRIzol LS (ThermoFischer, Waltham, MA, USA). RNA concentrations were determined by measuring the absorbance at 260 nm (A260) using a NanoDrop (ThermoFischer, Waltham, MA, USA). One microgram of total RNA was used for cDNA synthesis, using RevertAid H reverse transcriptase (ThermoFischer, Waltham, MA, USA), following the manufacturer’s instructions. Real time PCR was subsequently performed with a QuantStudio3 (ThermoFischer, Waltham, MA, USA) using the EvaGreen kit (Biotium, Fremont, CA, USA), according to the supplier’s instructions. Melting curves were analysed to ensure amplification specificity and absence of primer–dimer formation. The primers are listed in Appendix A. Primer PCR efficiencies were determined with a serial dilution of a mixed cDNA sample (1:2, 1:4, 1:8, and 1:16). All qPCR reactions were conducted as technical duplicates for each sample, and in total, three different silenced plants were analysed per TRV construct (three biological replicates). Water and no-template controls were included in each qPCR experiment as negative controls for each primer set. Cycle threshold (Ct) values were calculated using the auto baseline function in the QuantStudio software (ThermoFischer, Waltham, MA, USA). Duplicates for which the Ct value differed more than 0.5 were not considered and removed from the analysis. Finally, relative gene expression levels were calculated using the obtained Ct values in the software qBase+ (Biogazelle, Ghent, Belgium), applying a method that corrects for different primer efficiencies. Finally, the gene expression data were normalised using the *N. benthamiana APR* gene (NbS00020268g0004.1) as an internal reference [52].

## 3. Results

### 3.1. TSWV RNPs Isolated from N. benthamiana Contain Proteins Implicated in Plant Defence Responses and Solute Transport

As *N. benthamiana* is hypersusceptible to TSWV, viral RNPs isolated from this plant species are likely to contain host proteins crucial for the viral infection cycle, such as viral transcription, translation, assembly, and cell-to-cell movement. To identify associated host proteins, systemically infected leaf material of *N. benthamiana* was collected 14 days post TSWV inoculation (dpi), and the TSWV-containing RNPs were isolated by means of density-based ultra-centrifugation (Figure 1A) [29,33]. As control samples, similar cellular fractions were isolated from mock-inoculated plants. Immunoblotting confirmed that the plant TSWV RNP fraction contained both the viral L and N proteins (Figure 1B). In addition, immune electron microscopy revealed that the presence of nucleocapsid aggregates in the fractions (Appendix A). To confirm that the viral genomic segments were all present in these TSWV RNP samples, the purified fractions were rub-inoculated onto *N. benthamiana* leaves. As expected, the *N. benthamiana* plants inoculated with control samples (isolated from mock-treated plants) did not develop TSWV disease symptoms, while the plants inoculated with TSWV RNP samples did. To identify plant proteins present in these fractions, the samples were subjected to peptide mass fingerprinting (tryptic digestion followed by MS/MS analysis).

In total, 1802 proteins were identified, of which 1309 proteins were present in the RNP samples obtained from TSWV-infected plants (Figure 1C) (Appendix A). The TSWV RNP fraction contained 261 unique proteins (i.e., proteins only found in this fraction and not in the control), including all but one of the known viral proteins (Figure 1D), i.e., the C-terminal part of the viral glycoprotein (Gc) was not detected in any of the samples. Based on the LFQ values, the most abundant viral protein was the N protein, followed by the non-structural proteins NSm and NSs, respectively. In contrast, the N-terminal part of the viral precursor glycoprotein (Gn) and the L protein were both roughly 200-fold less abundant than NSm. In addition to unique proteins, 18 plant proteins were found to be significantly more abundant in the TSWV RNP fraction than in the mock control (Table 1). Intriguingly, five of these 18 proteins were predicted to be involved in plant immune responses, that is, Harpin-inducing protein 1-like 18 (NbS00006593g0102.1), Pleiotropic drug resistance protein 1 (NbS00010523g0001.1), Cysteine-rich receptor-like kinase 2 (NbS00022441g0017.1), Pathogenesis-related protein 1 (NbS00061216g0001.1), and Glucan endo-1,3-β-glucosidase (NbS00010129g0001.1). In addition, two sugar transporters (NbS00062251g0003.1 and NbS00058697g0004.1) and two inorganic phosphate transporters (NbS00006199g0011.1 and NbS00023594g0004.1) were more abundant in the TSWV RNP fraction than in the control. Furthermore, a DnaJ homolog subfamily B member 13 (DnaJB13/HSP40) (NbS00005708g0012.1) proved to be enriched. A close homolog of this DnaJ (HSP40) protein, NtDNaJ_M541 from *N. tabacum*, was previously reported to interact with the viral protein NSm [53]. In contrast, 493 proteins were unique for the mock samples, and 15 proteins were significantly more abundant in the mock than the TSWV RNP samples. Based on their annotation, the vast majority of the proteins enriched in the mock samples (13/15) were predicted to be localised in the chloroplast and involved in photosynthesis. This was in line with the colour (green) of the RNP pellets of the mock samples (Figure 1A). Together, these results indicate that, by purifying RNPs from TSWV-infected tissue, host proteins were co-purified with viral RNPs that are implicated in, amongst others, the defence responses and solute transport processes.

### 3.2. The TSWV RNP-Resident N. benthamiana Proteins Display an Overrepresentation for GO Terms Implicated in Immune System Process, Protein Folding, and Transport

To investigate whether the host proteins present in the TSWV RNP fractions are implicated in cellular processes important for viral assembly, replication, viral translation, and/or RNP cell-to-cell movement, a gene ontology (GO) enrichment test was conducted, focusing on biological processes (BP branch of the GO categories). To this end, a combined list of 266 plant proteins was compiled, which included both the unique (261) and (18) significantly accumulated plant proteins found in the TSWV RNP fractions. Because the genome of *N. benthamiana* remains poorly annotated to this date, we first identified for each *N. benthamiana* protein ID the closest homologue in the model plant Arabidopsis using BlastP. This resulted in a total of 244 best hits for Arabidopsis. As TSWV has a broad host range that includes Arabidopsis [54]*,* we anticipate that these homologues have a related conserved role in Arabidopsis for the TSWV disease cycle. This analysis revealed that the GO terms ‘*Response to stress*’ (GO: 0033554), with the subcategories ‘*Oxidative stress*’, ‘*Osmotic stress*’, and ‘*Defence responses*’, were significantly enriched (Figure 1E). In general, these stress responses are upregulated in response to various extracellular stimuli. Proteins implicated in plant immune responses were also enriched in the TSWV RNP fraction, in particular, the GO terms ‘*Immune system process*’ (GO: 0002376) and ‘*Cell death*’ (GO: 0008219). Examples of proteins falling in these two categories were ARGONAUTE 2 (AGO2), several receptor-like cytoplasmic kinases (RLCKs), and CALRETICULIN 3 (CRT3). Apparently, an antiviral immune response is mounted in the TSWV-infected cells [55,56,57].

Furthermore, proteins implicated in ‘*Protein folding*’ (GO: 0006457) and ‘*Transport*’ (GO: 0006810) were enriched in the TSWV RNPs. To demonstrate the specificity of this GO enrichment test, the same analysis was conducted for the mock fraction. This gave a list of 508 *N. benthamiana* proteins, of which 501 proteins had a clear ‘best hit’ in Arabidopsis. In this case, the GO analysis suggested that the mock fraction contained primarily proteins involved in cell division and chloroplast- and photosynthesis-related processes (Figure 1F). Together, these results suggest that, two weeks post TSWV inoculation*,* a defined set of *N. benthamiana* proteins had associated with viral RNPs, which were annotated as proteins involved in stress and plant defence responses, but also proteins that act in host processes that could be implicated in virus assembly and cell-to-cell movement of the virus [53,58,59,60].

Recently, a compendium of RNA-binding proteins from Arabidopsis was obtained using an affinity purification with poly-A resin combined after UV crosslinking of associated proteins to mRNA (so-called RBPome) [61]. This RBPome was shown to be enriched in proteins implicated in different mRNA processes. As we aimed to identify host factors involved in TSWV replication and translation, we examined whether the GO terms overrepresented for the TSWV RNPs were also enriched for this Arabidopsis RBPome, which comprises, in total, 1147 different proteins [61]. This GO enrichment test revealed that Arabidopsis RBPome is composed of proteins mostly implicated in RNA metabolic processes, RNA processing, translation, and Ribonucleoprotein complex biogenesis, thus encompassing all facets of RNA biogenesis, from synthesis to decay (Appendix A). Strikingly, two GO terms overlapped between the TSWV RNP-associated proteins and the Arabidopsis RBPome, namely ‘*Immune system process*’ and ‘*Protein folding*’. Overall, this GO analysis revealed that the TSWV RNPs were mostly obtained from plant cells where viral assembly was already advanced and/or viral latency was already prominent, while replication of the viral genome and viral translation was much reduced at the stage of leaf sampling.

### 3.3. TSWV RNPs Produced in Yeast Associate with Cellular Proteins Involved in Transcription and Translation

Even though bakers’ yeast is a non-host for TSWV, and yeast is apparently incompatible for viral transcription, yeast still supports replication of plasmid-encoded vcRNA segments into viral sense RNA (vRNA) [27]. Here, we assessed whether this artificial yeast-based TSWV replicon system [27] can be used to identify (unknown) yeast proteins that associate with ‘TSWV RNPs’, with the idea that many proteins involved in viral replication and translation are still conserved between yeast and Arabidopsis. The functional role of plant homologues of such yeast protein-coding genes can then be studied in planta during TSWV infection, albeit that, in plants, metabolic routes and signalling pathways are often more complex and suffer from gene redundancy.

To this end, replication-competent RNPs (rcRNPs) were reconstituted and purified from yeast expressing yeast codon-optimised variants of the *TSWV L* and *N* genes (N+L), together with an RNA molecule that mimics the genomic *S* segment of TSWV (*S*-RNA), hereafter called rcRNPs (*S*-RNA+N+L) (Figure 2A). To identify proteins that associated specifically with these rcRNPs, control fractions were also purified from yeast, i.e., (a) non-functional RNPs (nfRNPs) where TSWV *S*-RNA was expressed together with *TSWV N* (*S*-RNA+N) or (b) the *S*-RNA segment alone. Upon purification, presence of the viral L and N proteins in the RNP samples was verified by immunoblotting (Figure 2B), while the presence of the *S*-RNA was confirmed by RT-PCR (Figure 2B). The L protein was detected in two of three rcRNP samples. The rcRNP samples contained substantially increased levels of the N protein compared to the nfRNP samples. To confirm that the rcRNPs were intact upon purification, they were examined by electron microscopy after immunostaining of the N protein (Appendix A). Indeed, nucleocapsid aggregates were observed that were reminiscent of the aggregates seen for RNPs purified from TSWV-infected plants.

Using peptide mass fingerprinting, the protein composition of the collected yeast RNP fractions was determined. In total, 485 yeast proteins were identified in the different yeast samples (Appendix A). Of these, 447 proteins were identified in the rcRNPs (*S*-RNA+N+L). Among the 447 identified proteins, 165 proteins were unique for the rcRNP samples (Figure 2C). Fifty yeast proteins were present in both the rcRNP (*S*-RNA+N+L) and nfRNP (*S*-RNA+N) samples. In total, 295 and 248 proteins were identified in the nfRNPs and the samples expressing only the RNA *S* (*S*-RNA), respectively. Finally, 22 proteins proved to be unique for the nfRNPs in comparison to the ‘RNA *S* segment alone’ (*S*-RNA). Besides unique protein hits, we also examined whether some co-purifying proteins were significantly enriched in the nfRNPs compared to the ‘RNA *S* segment alone’. This did not yield any additional proteins. Three yeast proteins were specifically enriched in the rcRNPs (*S*-RNA+N+L), namely Ribosome biogenesis protein NSA2 (P40078), ATP-dependent RNA helicase MAK5 (P38112), and 40S Ribosomal protein S0-B (P46654) (Figure 2D). As these three proteins are implicated in transcription and translation, this suggests that expression of the L protein apparently resulted (in part) in recruitment of the host translational machinery to the TSWV genomic RNA.

### 3.4. The Yeast TSWV-like RNP Interactome Is Overrepresented for GO Terms Associated with Translation and Ribosome Biogenesis 

As both the viral proteins and the viral RNA segments can be expected to interact with cellular proteins involved in viral translation and replication, a GO enrichment analysis was performed of the yeast TSWV-like RNP interactome. First, we assessed whether certain GO terms were overrepresented for the 165 proteins that specifically co-purified with rcRNPs (*S*-RNA+N+L). In total, four GO categories were enriched, namely ‘*Single-organism metabolic process*’ (GO:0044710), ‘*Cellular process*’ (GO: 0009987), and ‘*Cellular component organization or biogenesis*’ (GO: 0071840), with one shared sub-category, ‘*Mitochondrial organization*’ (GO: 0007005) (Figure 2E). These processes are not per se implicated in viral replication and translation. The list of 22 unique yeast proteins in the nfRNPs (*S*-RNA+N) showed enrichment for two GO terms, namely ‘*Cellular component organization or biogenesis*’ (GO: 0071840) and one sub-category thereof, ‘*Ribosomal small subunit biogenesis*’ (GO: 0042274) (Figure 2F). In contrast, the set of 50 yeast proteins that were present in both the rcRNPs and nfRNPs was enriched for proteins implicated in transcription and protein translation, with an overrepresentation of the GO terms ‘*Ribosome biogenesis*’ (GO: 0022613), ‘*RNA processing*’ (GO: 0006396), and ‘*Macromolecular complex subunit organisation*’ (GO: 0043933) (Figure 2G). This indicates that the artificial TSWV yeast replicon system appears to recruit proteins associated with mRNA translation in yeast.

As a TSWV infection is likely to modulate host RNA biogenesis and mRNA translation, we determined whether the yeast-produced TSWV RNPs were enriched for yeast RNA-binding proteins. To this end, the overlap in overrepresented GO terms was determined between the viral RNPome (i.e., the combined list of 237 proteins identified in the rcRNPs and nfRNPs) and a published dataset of 671 yeast mRNA-binding proteins (yeast RBPome) [62]. The GO overrepresentation analysis of the yeast RBPome revealed that it is enriched for proteins implicated in RNA metabolic processes, transcription, translation, and ribosome biogenesis (Appendix A). When looking at the overlap between the yeast RBPome and the viral RNPome, both protein sets were enriched for processes involved in translation and ribosome biogenesis. Finally, two GO terms were unique for the viral RNPome, i.e., ‘*Protein folding*’ and ‘*Membrane fusion*’. The latter suggested that, at least to some extent, virus particle assembly had occurred in the yeast replicon system.

### 3.5. TSWV RNPomes Purified from Yeast and N. benthamiana Contain Conserved Proteins Potentially Involved in Viral Replication

While the rcRNPs retrieved from the TSWV yeast replicon system were enriched for proteins implicated in translation, the plant-derived TSWV RNPs were not. To determine whether universal cellular factors were present in both samples, we compared the functional orthology of the different proteins in the two lists (237 and 266 proteins, respectively) using the EggNOG V5.0 database [45], which is a database composed of gene evolutionary histories, orthology relationships, and functional annotations between organisms. The overlap in annotations between the two datasets revealed that 21 of the *N. benthamiana* proteins that co-purified with the TSWV RNPs, had a predicted orthologous relationship with 15 of the yeast proteins detected in the rcRNPs (Table 2). Interestingly, five plant proteins appear to have a common functional ancestor with the yeast protein Vba4 (Vacuolar basic amino acid transporter 4, Q04602), as they share as orthologous relationship ‘Intracellular trafficking, secretion, and vesicular transport’ (EuKaryotic Orthologous Groups (KOG)-0254). The annotation of the five Vba4-orthologous plant proteins suggested that they play a role in (sugar) transport. Furthermore, eight plant proteins were predicted to act in ‘Post-translational modification, protein turnover, and chaperones’, as described by different KOG IDs (KOG0019, KOG101, KOG191, KOG0714, KOG0715, KOG0730). Interestingly, the plant DnaJ homolog subfamily B member 13 (DnaJB13/HSP40) (NbS00005708g0012.1) has a shared orthologous relationship with the yeast protein SIS1, which is also annotated as a DnaJ/Hsp40 homologue (KOG0714). SIS1 is a yeast protein required for the initiation of mRNA translation [63]. Three plant proteins were annotated as ‘Translation, ribosomal structure and biogenesis’ and only one protein in ‘Replication, recombination and repair’. The latter two functional annotation groups are of interest, as they indicate universal proteins that potentially facilitate viral replication and/or transcription/translation.

### 3.6. Functional Studies on the TSWV RNP Interactome Mark an Isoform of the Eukaryotic Translation Elongation Factor 1A as a Pro-Viral Factor of TSWV

To interrogate whether any of the plant proteins that co-purified with the TSWV RNPs is important for viral replication *in planta*, candidate proteins were selected for further studies. As selection criteria, we used (i) proteins enriched in the plant RNP fraction, (ii) plant proteins that share an orthologous relationship with the identified yeast proteins, and (iii) proteins linked to processes involved in mRNA transcription or translation (Appendix A). To evaluate whether the 61 selected proteins have a role in viral replication, expression of the corresponding candidate genes in *N. benthamiana* was suppressed by means of virus-induced gene silencing (VIGS) [31]. Two weeks post agro-inoculation of the VIGS clones, the silenced plants were challenged with TSWV and development of TSWV disease symptoms was subsequently monitored in the weeks following (Figure 3). Silenced plants that developed normal TSWV symptoms (i.e., similar to the negative control TRV::*GUS*) were disregarded for further studies. Of the 61 silencing constructs tested, inoculation of four TRV constructs compromised TSWV symptom development to the levels attained with the positive control for VIGS-mediated loss of disease susceptibility (TRV::*TSWV N*). The VIGS constructs that resulted in compromised TSWV disease development contained gene fragments of the *Charged multivesicular body protein 1* (*CHMP1*; NbS00008317g0004.1), *β-1,3-glucanase* (*βGLU*; NbS00034922g0015.1), *Ammonium transporter 1* (*AMT1*; NbS00056501g0001.1), and a homologue of *eukaryotic translation elongation factor 1A* (*eEF1A*; NbS00023178g0001.1). Gene silencing with TRV::*CHMP1* and TRV::*AMT1* did not impact the development of *N. benthamiana*, i.e., the silenced plants showed no clear growth phenotype when compared to the negative control (TRV::*GUS*) (Figure 3A). Interestingly, silencing of *CHMP1, AMT1*, and *βGLU* resulted in partial resistance to TSWV, as the onset of TSWV disease symptoms was delayed, and the TSWV titres were reduced compared to the negative control (*GUS*-silenced plants), although they were never below the detection limit as seen for the positive control TRV::*TSWV N* (Figure 3B). Three weeks post TSWV inoculation, all plants silenced for *CHMP1, AMT1*, and *βGLU* showed, nevertheless, symptoms of a systemic TSWV infection. Agro-infiltration of TRV::*eEF1A* did not result in a uniform plant growth phenotype, i.e., some plants (8 out of 14 plants) developed normally, although they were slightly smaller than the negative control, whereas the others were severely stunted and had chlorotic leaves (5/14), and one plant had already collapsed soon after TRV inoculation (1/14). TSWV titres of the *eEF1A* silenced plants were reduced compared to our positive control (TRV::*TSWV N*). To confirm that gene silencing had taken place, the transcript levels of *CHMP1, AMT1, βGLU*, and *eEF1A* were determined using RT-qPCR 14 day post gene silencing. The transcript levels of the targeted genes suggest that they were each correctly silenced by the corresponding TRV clones in comparison to the negative control (TRV::*GUS*) (Figure 3C). Combined, this experiment shows that the *CHMP1, AMT1, βGLU,* and *eEF1A* gene products act as pro-viral factors for TSWV and that their knockdown in *N. benthamiana* indeed resulted in (partial) resistance to TSWV.

### 3.7. Silencing of One Specific Isoforms of the eEF1A Gene Family in N. benthamiana Is Sufficient for Loss of TSWV Susceptibility

As the TRV::*eEF1A* construct gave a range of developmental phenotypes, we interrogated the predicted genomic targets for this construct using the Solgenomics VIGS tool [48]. As expected, this TRV construct targets foremost the mRNA encoding the eEF1A isoform present in the TSWV RNPs (NbS00023178g0001.1 of the Niben0.4.4 database corresponds to *Niben101Scf07423g04011.1* in the Niben1.0.1 database used by https://solgenomics.net, accessed on 4 March 2021), with, in total, 239 predicted small-interfering RNAs (siRNAs) of 21-nucleotides length derived from the VIGS construct (Appendix A). TRV::*eEF1A* also targets two additional family members, *Niben101Scf04639g06007.1* and *Niben101Scf12941g01003.1*, albeit with a lower probability (44 and 2 predicted siRNAs, respectively). To reconstruct the evolution of the *eEF1A* gene family in the Solanaceae plant family, a gene tree was made that contained the family members found in *N. benthamiana, N. attenuata, Capsicum annuum*, *Solanum lycopersicum*, and *S. tuberosum* (Figure 4A). *eEF1A* homologues of Arabidopsis [64] and the monocot species *Oryza sativa* [65] were added as outgroups. Of note, all plant genomes inspected contained three to five *eEF1A* homologues, while *N. benthamiana* has eight homologues. In the *Solanaceae*, the *eEF1A* family is represented by two gene clades present in all species analysed, here named ‘clade A’ and ‘clade B’ [66]. The corresponding eEF1A family member for which the gene product was present in the TSWV RNP samples belongs to gene clade B, which includes one additional gene in *N. benthamiana* (Niben101Scf04639g06007.1). The VIGS construct is now renamed TRV::*eEF1A.B.* The two clade B paralogs in *N. benthamiana* have a nucleotide sequence similarity of 99%, suggesting that they result from a gene duplication that likely emerged from the polyploidisation event that shaped the allotetraploid *N. benthamiana*. In contrast, clade A contains six genes, each sharing around 92% sequence similarity with the genes present in clade B.

As the *eEF1A* clades A and B share high sequence similarity with each other, we wondered whether clade A is also required for TSWV disease susceptibility or if only clade B is important for completing the TSWV infection cycle. To that end, two additional, more specific TRV constructs were generated. One construct targeted only two genes of clade A (*Niben101Scf08618g01001.1* and *Niben101Scf08618g01002.1*) (TRV::*eEF1A.A1*), while the other targeted all six gene members of clade A (TRV::*eEF1A.A2*) (Appendix A). Agro-inoculation of TRV::*eEF1A.A2* in *N. benthamiana* resulted in strong necrosis, with all plants collapsing within two weeks after agro-inoculation (n=14 plants) (Figure 4B). Therefore, we discontinued working with this construct. In contrast, agro-inoculation of TRV::*eEF1A.A1* resulted only in a mild developmental phenotype, almost comparable to the negative control (TRV::*GUS*). Apparently, one or more members of clade A are essential for plant fitness.

Gene expression analyses showed that the combined transcript levels of the two genes for *eEF1A.A1* and *eEF1A.B* were reduced in TRV::*eEF1A.A1* and TRV::*eEF1A.B* plants, respectively, compared to the control (TRV::*GUS*) (Figure 4C). In addition, no cross-silencing was observed for the two genes of clade A by TRV::*eEF1A.B* or for the two genes belonging to clade B by TRV::*eEF1A.A1.* These silenced plants were then challenged with TSWV at two-weeks post TRV agro-inoculation. All TRV::*eEF1A.A1* inoculated plants (n = 7) developed clear TSWV disease symptoms within two weeks, similar to the negative control (TRV::*GUS*). The observed disease symptoms were confirmed by ELISA against the viral N protein and showed that TSWV titres were similar between TRV::*eEF1A.A1*- and TRV::*GUS*-inoculated plants (Figure 4D). Thus, the two genes targeted by TRV::*eEF1A.A1* at the same time are not essential for TSWV disease development. As the original *eEF1A* silencing construct, targeting foremost clade B, reduced TSWV infectivity and the construct TRV::*eEF1A.A1* targets members of clade A, these data argue that the two paralogs of clade B acts as pro-viral factors for TSWV susceptibility.

## 4. Discussion

In the past, TSWV RNPs have been purified from *Nicotiana rustica* to study viral RNA synthesis and particle morphogenesis and for diagnostics purposes [12,24,25,33]. To our knowledge, RNPs have not yet been used as a biological source to identify host factors involved in TSWV replication and/or translation. Our analysis of the protein composition of TSWV RNPs has successfully yielded four novel host factors, *AMT1, βGLU, CHMP1*, and *eEF1α,* that, upon gene silencing in *N. benthamiana*, convey reduced susceptibility to TSWV.

In particular, we found that one specific isoform (‘clade B’) of the eEF1A protein family is a pro-viral factor for TSWV. Silencing of the corresponding two genes resulted in reduced TSWV titres in *N. benthamiana*. This finding is consistent with an earlier report that the eEF1A protein family is required for in vitro TSWV transcription and facilitates replication of TSWV RNA [25]. eEF1A is an abundant cellular protein that binds and delivers amino-acylated (aa-)tRNAs to ribosomes during the mRNA translation (elongation) process [67]. Moreover, the eEF1A protein family shows a high degree of sequence conservation across the different kingdoms of life. Proteins of other plant viruses also interact with different eEF1A isoforms, e.g., *Tomato bushy stunt virus* (TBSV), *Tobacco mosaic virus* (TMV), and *Turnip yellow mosaic virus* (TYMV) [68,69,70,71,72]. Besides its role in translation, eEF1A interacts with filamentous actin, impacts nucleocytoplasmic travelling of RNA, and it controls the turnover of damaged proteins for degradation and regulates apoptosis and the response to a heat shock [72,73,74,75,76]. Possibly, eEF1A is used differently during the various stages of the TSWV infection cycle. The notion that silencing *eEF1A* family members would potentially affect protein translation can be dismissed, as others showed that silencing of *eEF1A* does not affect the global transcription or translation activity in *N. benthamiana* [66].

Our phylogenetic analysis of the eEF1A protein family revealed two clades in *Solanaceae* (Figure 4). Only silencing of the two genes that belong to clade B suppressed TSWV disease development and reduced the TSWV viral titres in *N. benthamiana*. In contrast, knockdown of the six genes that together form clade A with a single TRV construct (TRV::*eEF1A.A2*) resulted in collapse of the silenced plants. These data argue that the *eEF1A* clade A genes are together essential for plant development. The limited growth penalty seen for TRV::*eEF1A.A1*, in comparison to TRV::*eEF1A.A2*, suggests the existence of gene redundancy in clade A, or that the remaining four genes are essential in combination. Gene redundancy would also explain why inoculation with the TRV::*eEF1A.A1* construct did not suppress TSWV infectivity. Others reported that silencing of the *eEF1As* of the B clade in *N. benthamiana* causes severe stunting, similar to what was observed with the TRV::*eEF1A.B* construct designed here [66]. Future studies should determine whether there are functional differences between the two *eEF1A* isoform clades of the *Solanaceae*, in particular, in their role as pro-viral factors for different plant viruses. Moreover, it will be interesting to investigate whether loss-of-function mutations of the closest homologues of the *eEF1A* B clade in other *Solanaceous* species confers reduced disease susceptibility towards TSWV.

Besides *eEF1A*, three other genes, *βGLU, CHMP1*, and *AMT1* were found to impact TSWV infectivity, as silencing of these genes also resulted in partial TSWV resistance. β-1,3-glucanase (βGLU) is member of a protein family that can hydrolyse β-1,3-glucan polysaccharides. As such, β-1,3-glucanases have anti-fungal properties, as they are able to degrade β-1,3-glucans present in fungal cell walls [77,78,79]. Moreover, 1,3-glucans are found in the form of callose in plant cell walls and around plasmodesmata [80]. The removal of callose from the plasmodesmata by the β-1,3-glucanase enzyme activity increases the size exclusion limit, thereby enabling larger molecules to move from cell to cell. Hypothetically, by silencing of *βGLU*, the overall enzymatic activity of βGLU will be reduced, leading to an increased callose deposition at the plasmodesmata. As a consequence, the size exclusion limit of the plasmodesmata will decrease, thereby preventing the cell-to-cell movement of TSWV RNPs. This is supported by a study reporting that the upregulation of *AtβGLU* (At3g57260) from Arabidopsis during a TSWV infection, correlated with increased TSWV titres in systemically infected leaves [81]. In the same study, silencing of *AtβGLU* in Arabidopsis resulted indeed in reduced TSWV infectivity. The role of β-1,3-glucanases in viral pathogenesis was also shown for TMV and *Tobacco necrosis virus* (TNV) in, respectively, *N. tabacum* and *N. sylvestris* knockdown mutants of the *β-1,3-glucanase* gene family [82]. In this study, inoculation of such mutants with TMV revealed decreased susceptibility, as observed by the (i) formation of smaller and less TMV lesions, (ii) reduced TMV titres, and (iii) increased callose deposition around the plasmodesmata. Taken together, our work revealed that, by analysing host proteins that associate with TSWV RNPs, a pro-viral factor that potentially facilitates the cell-to-cell movement of this virus in *N. benthamiana* has been identified.

CHMP1 is a subunit of the endosomal sorting complex required for transport (ESCRT)-III pathway, which is involved in membrane remodelling and release of invaginated endosomes from multivesicular bodies for degradation in the vacuole/lysosome [83]. Besides playing a role in endosomal sorting, the ESCRT-III machinery is also involved in autophagy-mediated degradation of chloroplasts (chlorophagy) [83,84]. In addition, this machinery is used during viral replication, as, for example, *Tomato bushy stunt tombusvirus* (TBSV) recruits different ESCRT subunits for the formation of the viral replication compartment at the peroxisome to evade antiviral immune responses [85]. A similar role for CHMP1 in creating a safe environment for viral replication could be perceived for TSWV, yet the exact function of CHMP1 in TSWV susceptibility needs further study.

For AMT1, we showed that *N. benthamiana* plants silenced for the encoding gene developed normally yet exhibited a clear reduction in their susceptibility towards TSWV. In Arabidopsis, it was shown that AMT1.1 functions as a negative regular of plant defence, and *AMT1.1* mutant lines exhibited a constitutive defence response, resulting in increased resistance to the necrotrophic fungal pathogen *Plectosphaerella cucumerina* and the biotrophic bacterium *Pseudomonas syringae* [86]. The role of AMT1 as a negative regulator of defence might explain the reduced susceptibility to TSWV when silencing the corresponding gene. More research will be required to further unravel the role of AMT1 in the TSWV infection cycle.

The DnaJ homologue subfamily B member 13 (DnaJB13) was also associating with the TSWV RNPs. We also isolated a homologous protein from the yeast replicon system, SIS1 (a DnaJ/HSP40 family member). Silencing of *DnaJB13* in *N. benthamiana* had no apparent effect on TSWV infectivity, as the TRV::*DnaJB13* plants became equally infected by TSWV as the negative control (TRV::*GUS*). DnaJ/HSP40 family members from *N. tabacum*, *S. lycopersicum* and Arabidopsis were reported before to interact with the TSWV movement protein NSm, and therefore to play a putative role in viral movement [53,59]. Besides the recruitment of DnaJ homologues by TSWV, other viruses like *Potato virus X* (PVX)*, Potato virus Y* (PVY) and TMV, were shown to directly interact with DnaJ homologues [60,87,88]. The DnaJ protein family forms a diverse subset of molecular chaperones that assists in protein folding and macromolecular protein assembly by acting as co-chaperones of the HSP90-HSP70-HSP40 complex [89,90]. In addition, DnaJ proteins assist in mRNA translation and protein degradation and play a role in virus infection in eukaryotes [60,91]. For instance, SIS1 was shown to be associated with the 40S ribosomal subunits and it is proposed to act in translation initiation [63]. Taken together, our findings further support the recruitment of DnaJ homologous proteins by TSWV, yet their role during TSWV infection will be the subject of future studies.

Besides identifying host factors from plants, we also investigated potential conserved host factors involved in viral RNA replication using the TSWV replicon system in yeast. We identified that 15 yeast factors have functional orthology with 21 host factors from *N. benthamiana*. Subsequent validation of these *N. benthamiana* host factors and their relation in TSWV pathogenicity did not lead to reduced viral susceptibility. One explanation for no S gene being identified via this approach may be the allotetraploid genome of *N. benthamiana,* which consequently results in numerous gene copies. The observation that, among the yeast–*N. benthamiana*-selected candidate genes, no S genes were identified does not imply that these are not required for TSWV pathogenicity.

We observed that the purified TSWV RNPs from *N. benthamiana* contained plant proteins implicated in vesicle formation and protein transport. Potentially, this signifies that the time points chosen for collecting the leaf material were too late in the infection process, and that mature virus particles were already assembled and stored at the Golgi apparatus [92]. Clearly, it will be interesting to isolate RNPs from plant cells at different time points to gain more insight in the different steps of the genome replication/transcription, but it will be imperative to develop a system where the TSWV infection cycle can be synchronised in the infected plant cells. The closest to such a system would be RNP transfections of protoplasts. In contrast to TSWV RNPs purified from *N. benthamiana*, we observed that the yeast RNPs contained proteins involved in translation and ribosome biogenesis. These processes are expected to occur during the early stages of viral replication. To obtain a better understanding of the different host factors associated with RNPs, viral RNPs should also be isolated from other hosts, *e.g.,* thrips (*F. occidentalis*), Arabidopsis, or a crop plant, such as tomato. In conclusion, we here present a method that can be readily applied to other negative-stranded RNA viruses to identify important host proteins playing a role in susceptibility to viral diseases. 

## Figures and Tables

**Figure 1 viruses-13-02190-f001:**
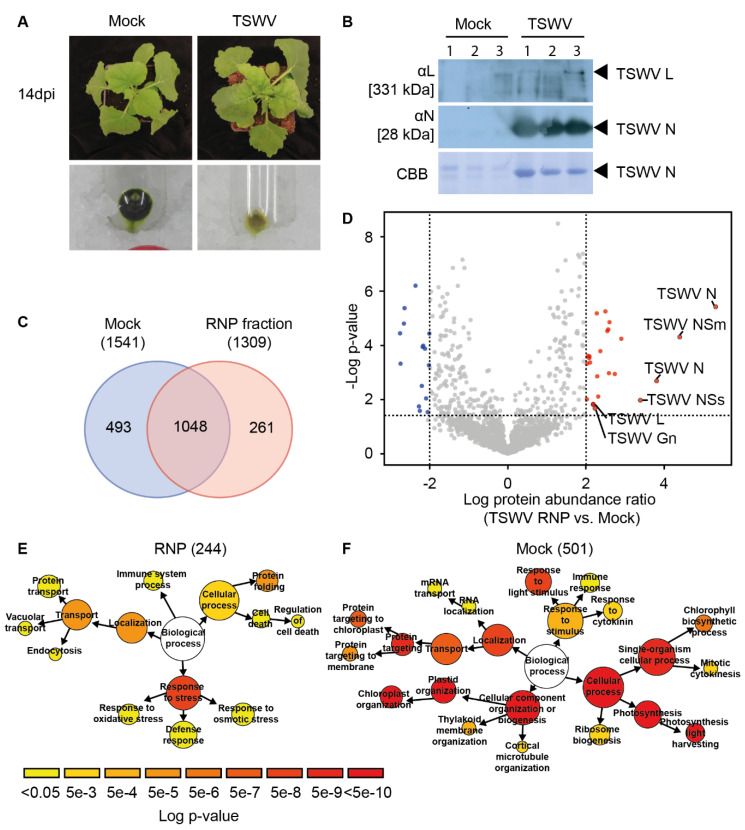
Mass spectrometry analyses of the TSWV RNP-containing fraction from *N. benthamiana*. (**A**) Ribonucleoprotein (RNP)-containing fractions were purified from TSWV-inoculated *N. benthamiana* and mock plants at 14 dpi. (**B**) Western blots of the RNP fractions purified from leaf material via density separation. For all three biological replicates of the RNP samples and the mock control samples, the presence of the viral L and N proteins was confirmed. (**C**) Venn diagram showing the overlap in proteins present in the RNP and mock samples. (**D**) Volcano plot of the proteins identified in the RNP samples compared to the mock control. All significant proteins are visualized by blue or red dots. All the viral proteins identified are indicated. (**E**,**F**) Gene ontology (GO) enrichment analysis of the plant proteins present in the (**E**) TSWV RNPs or (**F**) mock samples. The number of proteins per GO category is represented by the size of the circles. Scale bar indicates the log_10_ *p*-value cut-off values (hypergeometric enrichment test), with yellow and red indicating low and high level of significance, respectively.

**Figure 2 viruses-13-02190-f002:**
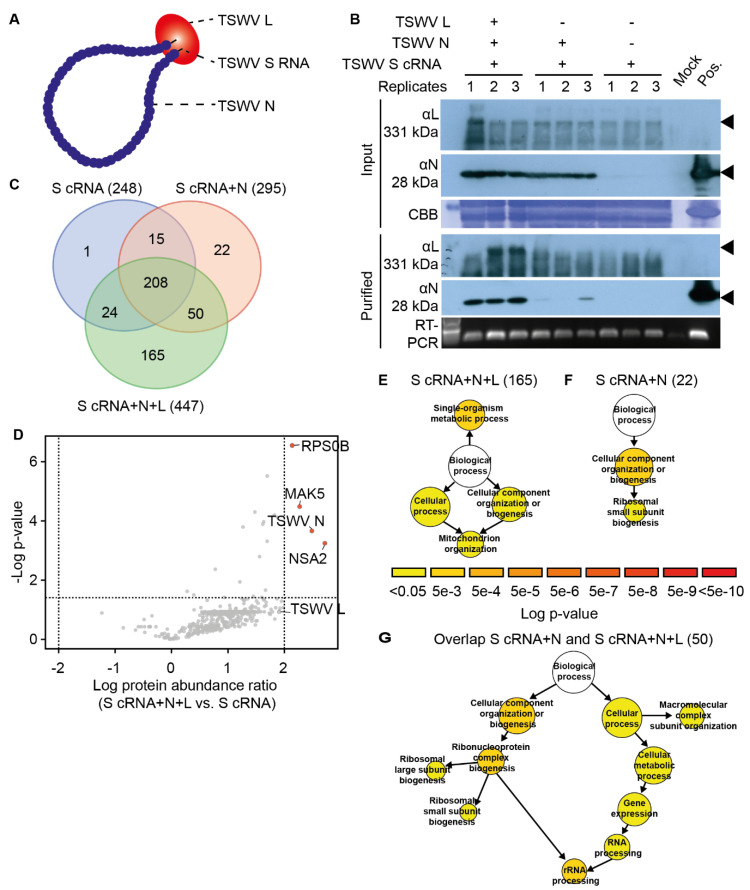
TSWV RNP expression in yeast and analysis of the TSWV RNP-containing fraction from yeast by mass spectrometry. (**A**) Schematic diagram of a single TSWV Ribonucleoprotein (RNP) particle consisting of the TSWV *S*-RNA genomic segment (black line) and the TSWV L and N proteins (red and blue circles, respectively). (**B**) Immunoblot analysis and DNA agarose gel (RT-PCR) depicting the three independent RNP fractions (top: replicates) isolated from yeast via density separation. For all three replicates of the rcRNPs and the two negative controls, nfRNPs (*S*-RNA+N) and *S*-RNA alone, the presence of the viral L and N proteins was determined, as well as the viral RNA; αL, polyclonal antibody raised against TSWV L protein; αN, polyclonal antibody raised against TSWV N protein; CBB, Coomassie Brilliant Blue gel staining to reveal equal protein loading. Presence of the viral *S*-RNA segment was confirmed using RT-PCR on cDNA. (**C**) Venn diagram depicting the overlap in yeast proteins identified by mass spectrometry in the three RNP samples. (**D**) Volcano plot showing significantly enriched proteins (Student’s t-test, *p*-value <0.05) in the rcRNPs compared to the *S*-RNA control. (**E**–**G**) Gene ontology (GO) enrichment analysis of the yeast proteins unique/enriched for the (**E**) rcRNPs, (**F**) nfRNPs, and (**G**) present in both the rcRNPs and nfRNPs. The number of proteins per GO category is represented by the size of the circles. Scale bar indicates the log_10_ *p*-value cut-offs (hypergeometric enrichment test), where yellow and red indicate low and high significance, respectively.

**Figure 3 viruses-13-02190-f003:**
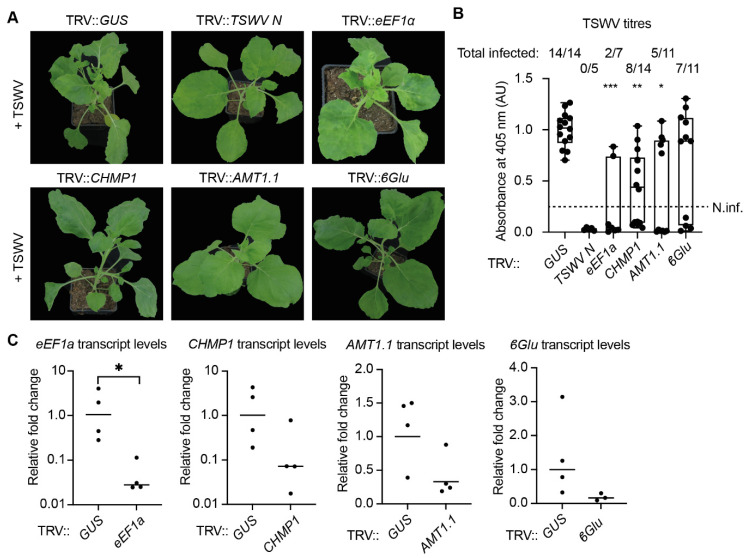
Knockdown of the genes *eEF1A*, *CHMP1*, *AMT1.1*, and *β**Glu* results in reduced TSWV titres in *N. benthamiana*. (**A**) Growth phenotypes of *N. benthamiana* plants silenced for *eEF1A*, *CHMP1*, *AMT1.1*, or *B-Glu*. Four-week-old silenced plants were rub-inoculated with TSWV, and pictures were taken two weeks later. (**B**) TSWV titres determined by DAS-ELISA of the TSWV-inoculated plants shown in (**A**); viral titres were normalised to negative control (TRV::*GUS*). Plants were considered to be non-infected when the viral titres were below 0.25 arbitrary units (AU), and the number of infected plants is indicated above the bars. Kruskal–Wallis test with a Dunn’s post hoc test was performed (*, *p*-value < 0.05; **, *p*-value < 0.01; ***, *p*-value < 0.001). (**C**) Transcript levels of *eEF1A*, *CHMP1*, *AMT1.1*, and *β**Glu* of the silenced plants shown in (**A**). Transcript levels were compared to the control plants (TRV::*GUS*) (n = 4). Unpaired Student’s *t*-test (*, *p*-value < 0.05).

**Figure 4 viruses-13-02190-f004:**
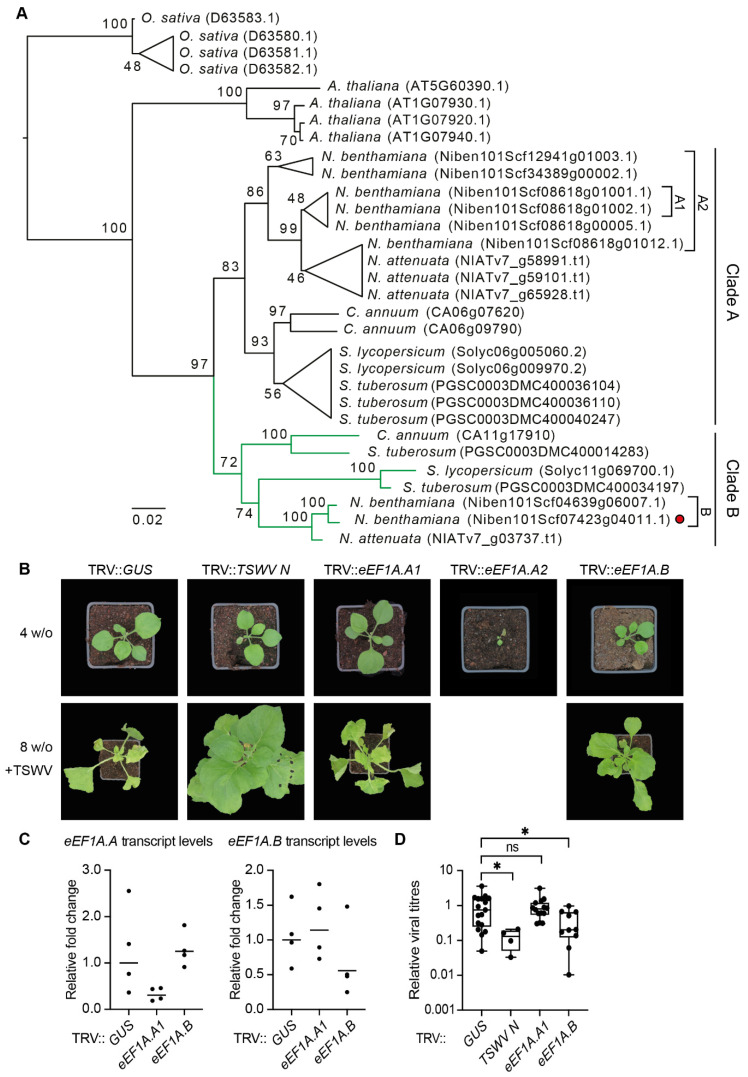
Solanaceous *eEF1A* isoforms cluster in two clades, clade A and B, of which clade B is required for TSWV susceptibility. (**A**) Phylogenetic analysis of the *eEF1A* coding sequences retrieved from Arabidopsis*, Capsicum annuum, Nicotiana attenuata, N. benthamiana*, *Oryza sativa, Solanum lycopersicum*, and *S. tuberosum*. The red dot indicates the *eEF1A* homologue of *N. benthamiana* that co-purified with viral RNPs. A1, A2, and B denote the VIGS constructs targeting the different *N. benthamiana eEF1A* genes. (**B**) Growth phenotype of *N. benthamiana* upon VIGS of *eEF1A.A1*, *eEF1A.A2*, or *eEF1A.B*, using TRV two weeks post agro-inoculation of the VIGS construct. (**C**). *eEF1*A transcript levels of the plants shown in (**B**) (n = 4). Unpaired Student’s t-test (*, *p*-value < 0.01). (**D**) DAS-ELISA showing the TSWV viral titres of plants shown in (**B**), at two weeks post TSWV inoculation.

**Table 1 viruses-13-02190-t001:** List of protein either enriched in the RNP fraction or the mock fraction. Proteins in bold are TSWV proteins.

**Proteins Significant Enriched in the RNP Fraction**				
**Protein Annotation**	**Gene ID**	**Log Protein** **Abundance Ratio**	**−Log *p*-Value**	**GO Term Biological Processes**
**TSWV N**	**P25999**	**5,322364807**	**5,418489703**	**Viral nucleocapsid**
**TSWV NSm**	**P36292**	**4,403109868**	**4,299333733**	**Host–virus interaction, transport**
Harpin inducing prot 1-like 18	NbS00006593g0102.1	2,500370344	5,248621579	Response to stress
Pleiotropic drug resistance protein 1	NbS00010523g0001.1	2,279739062	5,176989917	Response to stimulus, Transport
Probable mitochondrial chaperone BCS1-B	NbS00033465g0001.1	2,604620616	4,849324623	Response to stress
Cysteine-rich receptor-like protein kinase 2	NbS00022441g0017.1	2,573195457	4,582542858	Cellular metabolic process
Cytochrome b561	NbS00002670g0023.1	2,907094002	4,236962944	Response to stimulus
Sugar transporter 13	NbS00062251g0003.1	2,54924806	4,523826123	Transport
**TSWV N**	**Q0PTG4**	**3,811312199**	**2,677940859**	**Viral nucleocapsid**
Catalase isozyme 1	NbS00013764g0007.1	2,370932102	3,778977343	Response to stress
Pathogenesis-related protein 1	NbS00061216g0001.1	2,093765418	3,584313513	Defence response
Calcium-dependent lipid-binding (CaLB domain) family protein	NbS00018510g0001.1	2,740815957	2,934327942	Defence response
Glucan endo-1,3-beta-glucosidase	NbS00010129g0001.1	2,055405935	3,606080883	Response to stimulus, Transport
Inorganic phosphate transporter	NbS00006199g0011.1	2,083080928	3,522238151	Transport
Hexose transporter	NbS00058697g0004.1	2,590578238	2,964279133	Transport
DNA-J homolog 13	NbS00005708g0012.1	2,106231213	3,357389199	Protein folding
**TSWV NSS**	**P26002**	**3,394036452**	**1,964366825**	**Suppressor of RNA silencing**
Clathrin light chain 2	NbS00057576g0008.1	2,03093799	3,317251501	Establishment of localisation
ABC transporter B family member 21	NbS00020002g0005.1	2,291292191	2,847364381	Transport
Embryo-specific protein ATS3B	NbS00003054g0022.1	2,315136909	2,099199888	--
Protein SSUH2 homolog isoform x2	NbS00054171g0002.1	2,026340644	1,999542856	Protein folding
**TSWV L**	**P28976**	**2,189735095**	**1,828212439**	**RNA-directed RNA polymerase**
Inorganic phosphate transporter	NbS00023594g0004.1	2,184769948	1,807216819	Transport
**TSWV Gn**	**P36291**	**2,234362761**	**1,663396874**	**Host–virus interaction**
**Proteins Significant Enriched in the Mock Fraction**				
**Protein Annotation**	**Gene ID**	**Log Protein** **Abundance Ratio**	**−Log *p*−Value**	**GO Term Biological Processes**
Chloroplast photosystem I reaction centre V	NbC24101312g0001.1	−2,362002055	6,199989333	Photosynthesis
Photosystem I reaction centre subunit IV A	NbS00038432g0004.1	−2,64222304	5,36905629	Photosynthesis
Chlorophyll a-b binding protein 21	NbS00014580g0003.1	−2,658362707	4,797954045	Photosynthesis
Mg protoporphyrin IX chelatase	NbS00040680g0007.1	−2,756849289	4,437632643	Chlorophyll biosynthetic process
Root phototropism protein 2	NbS00008322g0011.1	−2,00761652	4,429897003	Phototropism
Glutamyl-tRNA reductase 1	NbS00054987g0005.1	−2,16445907	3,97925492	Chlorophyll biosynthetic process
DEAD-box ATP-dependent RNA helicase 3	NbS00021398g0012.1	−2,179090659	3,935555191	Ribosome biogenesis
Photosystem I reaction centre subunit IV B	NbS00019085g0006.1	−2,744261424	3,31588844	Photosynthesis
Photosystem I reaction centre subunit XI	NbS00028915g0014.1	−2,117512862	3,870465944	Photosynthesis
Suppressor of thylakoid formation 1	NbS00012196g0002.1	−2,017049789	3,25883739	Chloroplast organisation
Thylakoid lumen protein 18.3	NbS00009383g0013.1	−2,200131734	2,492686209	Chloroplast organisation
Chlorophyll a-b binding protein 13	NbS00020253g0009.1	−2,090600967	2,031389612	Photosynthesis
Protein plastid transcriptionally active 16	NbS00042661g0005.1	−2,273987929	1,735711971	Circadian rhythm
Pentatricopeptide repeat-containing protein	NbS00042677g0002.1	−2,25110515	1,569628025	Circadian rhythm
Oxygen-evolving enhancer protein 1	NbS00019818g0001.1	−2,049469153	1,523308383	Photosynthesis

**Table 2 viruses-13-02190-t002:** Overlap in functional annotations of proteins present in the RNP containing fraction of *N. benthamiana* and yeast.

*N. benthamiana* ID	Yeast ID	Gene Name (Yeast)	Protein Annotation	KOG Database	Function Annotation
NbS00019858g0012.1	P30624	FAA1	Long-chain-fatty-acid–CoA ligase 1	KOG1180	Lipid transport and metabolism
NbS00027624g0003.1	P38929	PMC1	Calcium-transporting ATPase 2	KOG0204	Inorganic ion transport and metabolism
NbS00037522g0005.1	Q06698	YLR419W	Putative ATP-dependent RNA helicase	KOG0920	Replication, recombination, and repair
NbS00036611g0008.1	P32481	GCD11	Eukaryotic translation initiation factor 2 γ	KOG0466	Translation, ribosomal structure, and biogenesis
NbS00009889g0007.1	P35723	YET1	Endoplasmic reticulum transmembrane protein 1	KOG1962	Translation, ribosomal structure, and biogenesis
NbS00009856g0011.1	P36520	MRPL10	54S ribosomal protein L10	KOG0846	Translation, ribosomal structure, and biogenesis
NbS00003478g0004.1	P46367	ALD4	Potassium-activated aldehyde dehydrogenase	KOG2450	Energy production and conversion
NbS00048826g0001.1	P32610	VMA8	V-type proton ATPase subunit D	KOG1647	Energy production and conversion
NbS00039262g0004.1	P40557	EPS1	ER-retained PMA1-suppressing protein 1	KOG0191	Energy production and conversion, post-translational modification, protein turnover, and chaperones
NbS00011481g0021.1	P15108	HSC82	ATP-dependent molecular chaperone	KOG0019	Post-translational modification, protein turnover, and chaperones
NbS00023686g0020.1	P25694	CDC48	Cell division control protein 48	KOG0730	Post-translational modification, protein turnover, and chaperones
NbS00015581g0008.1	P40358	JEM1	DnaJ-like chaperone	KOG0715	Post-translational modification, protein turnover, and chaperones
NbS00005708g0012.1	P25294	SIS1	SIt4 Suppressor1	KOG0714	Post-translational modification, protein turnover, and chaperones
NbS00006769g0021.1	P25294	SIS1	SIt4 Suppressor1	KOG0714	Post-translational modification, protein turnover, and chaperones
NbS00023331g0003.1	P10591	SSA1	Heat shock protein SSA1	KOG0101	Post-translational modification, protein turnover, and chaperones
NbS00040361g0003.1	P10591	SSA1	Heat shock protein SSA1	KOG0101	Post-translational modification, protein turnover, and chaperones
NbS00004424g0012.1	Q04602	VBA4	Vacuolar basic amino acid transporter 4	KOG0254	Intracellular trafficking, secretion, and vesicular transport
NbS00007660g0008.1	Q04602	VBA4	Vacuolar basic amino acid transporter 4	KOG0254	Intracellular trafficking, secretion, and vesicular transport
NbS00023311g0013.1	Q04602	VBA4	Vacuolar basic amino acid transporter 4	KOG0254	Intracellular trafficking, secretion, and vesicular transport
NbS00058697g0004.1	Q04602	VBA4	Vacuolar basic amino acid transporter 4	KOG0254	Intracellular trafficking, secretion, and vesicular transport
NbS00062251g0003.1	Q04602	VBA4	Vacuolar basic amino acid transporter 4	KOG0254	Intracellular trafficking, secretion, and vesicular transport

## Data Availability

The mass spectrometry proteomics data have been deposited to the ProteomeXchange Consortium via the PRIDE [42] partner repository with the dataset identifier PXD026246.

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
