# Peer review of "An Isoform of the Eukaryotic Translation Elongation Factor 1A (eEF1a) Acts as a Pro-Viral Factor Required for Tomato Spotted Wilt Virus Disease in Nicotiana benthamiana"

_viruses, 2021, doi:10.3390/v13112190_

Round 1

Reviewer 1 Report

In the manuscript by Helderman et al., the authors present the results of an investigation aimed at  addressing a knowledge gap on host factors supporting the processes of replication and transcription of an economically important virus, the tospovirus Tomato spotted wilt virus. As approach in the study, ribonucleoprotein complexes were purified from Nicotiana benthamiana plants and yeast, a non-host for this virus, but earlier successfully employed to establish a TSWV replicon system. Proteomics analysis and parallel comparison of the two systems allowed the authors to identify candidates factors playing a role in the TSWV infection. To further investigate this observation, experiments involving VIGS of candidate genes and challenging of the plants with TSWV were set up, and allowed to identify a eEF1a isoform (belonging to the B clade, as defined by the authors) as a pro-viral factor for TSWV disease. The results are consistent with early reports on the role of eEF1A to support TSWV RNA synthesis. In this work, the system TSWV-N. benthamiana has been considered and a VIGS approach, including othere candidates, has been further used to provide evidence of the (putative) role of factors in the TSWV infection cycle. Overall the presented method, combining a plant and yeast system, offered the possibility to integrate the data from different organisms, and indeed represent a valid approach. The manuscript is well written and presented. Further the authors clearly state the necessity of further studies to clearly understand the role of specific factors in TSWV susceptibility, and do not run into far-too-hypothetical considerations, which can sometime easily come up from -omics approaches. Rather, the results are presented considering the potential of future investigations, for confirmation with other, and more target-focused, approaches. As well they clearly point to future studies in insects and in crop plants, which indeed would be of extreme interest.  

I have only minimal comments and I'm supportive to consider the manuscript for publication in the Journal.      

Minor comments

-Keywords: many proteomics approaches are available nowadays; I would be more precise on the method used.

-L82: complementary

-Table1: the GO category could be added in the Table, to provide a classification of the protein, rather than a pure list.

-L507: focusing

-L669: "which actually does not take place"; can you please clarify considering what reported in L568: "still supporting the replication..". Please explain more profusely.

-Fig.3: provide information on the timing for the pictures presented in panel A. 

Author Response

Reviewer1:

Minor comments

-Keywords: many proteomics approaches are available nowadays; I would be more precise on the method used.
We agree with the referee and added the keywords: ‘NanoLC-MS/MS’ and further specified the keyword ‘proteomics’ into ‘SDS-PAGE based proteomics’.

-L82: complementary
We added the suggestion, resulting in: ‘viral complementary genomic S segment (vcRNA)’. 

-Table1: the GO category could be added in the Table, to provide a classification of the protein, rather than a pure list.
Thanks for this suggestion. The GO terms have now been added to the table as suggested.

-L507: focusing
We corrected the typo. Thanks for spotting it.

-L669: "which actually does not take place"; can you please clarify considering what reported in L568: "still supporting the replication..". Please explain more profusely.
The artificial RNP replicon system in yeast is capable of orchestrating viral genomic replication, from the RNA that originates from the yeast plasmid to vcRNA at least. However, viral transcription does not commence from this vcRNA and therefore viral mRNAs are not formed for protein translation. Interestingly, we did found yeast proteins that co-purified with the viral RNPs from the yeast cells that are (indirectly) implicated in protein translational based on their annotation. We have revised to line 653-655 to clarify this point in more detail.

-Fig.3: provide information on the timing for the pictures presented in panel A. 
Timing of the pictures shown is now indicated in the caption of Figure 3A.

Reviewer 2 Report

The paper deals with identifying the host factors required for TSWV infection. The authors conducted two types of screening, one using plants and the other using yeast, but the yeast screening did not seem to be valuable information because it did not affect the final identification of host factors. Although the overall description was carefully written, it was somewhat lengthy and there were some descriptions that seemed unnecessary or over-interpreted.
I evaluate this study as interesting, but  a substantial revision is needed to make this manuscript suitable for publication.
Some of the issues to be considered are commented below.

L58: If S genes for tospovirus have been reported in previous studies, it should be explained.
L439: Please move the arrows in Figure S1 so that the tip of each arrow points to the nucleocapsid aggregates.
L487: Please replace “factions” with “fractions”.
L507: Please replace “focussing” with “focusing”.
L536-540: Please provide evidence that the GO-enriched terms obtained in this section is related to virus assembly and cell-to-cell movement, or else discuss this in the Discussion.
L541-563, L648-670: The comparison with previous studies of RBPome is lengthy and unclear in its purpose and should be eliminated.
L560: What were the properties of the RNPs you analyzed? Viral particles, viral replication complexes, or something else? If they were viral particles, were the conditions of this experiment suitable for analyzing what you really wanted to analyze?
L593-599: In this experiment, we cannot know if the function of the purified RNPs was intact or not. You need to show that the purified RNPs have replication activity in vitro, or the phrase needs to be changed.
L645-647: It is necessary to discuss the meaning that proteins associated with mRNA translation were co-purified with RNPs, even though the yeast replicon system supports only viral replication, but not transcription and translation of viral mRNA. It is also necessary to discuss the reason why candidate S genes were not obtained in this experiment.
L812-813: Please replace “VIGS construct construct” with “VIGS construct”.
Table 1: Please change the commas to periods in log value and reduce the number of digits to three.

Author Response

L58: If S genes for tospovirus have been reported in previous studies, it should be explained.
We agree with the referee that this specific topic was not addressed in the introduction. We have now three references that report the identification of candidate S genes for TSWV (L60-L62). The referred S genes are mutant versions of the 9-lipoxygenase (9LOX) and a-dioxygenases (a-DOX) in Arabidopsis (Garcia-Marcos et al., 2013), root hair defective 3 (rhd3) also in Arabidopsis (Feng et al., 2016), and suppressor of the G2 allele of skp1 (SGT1) in N. benthamiana (Quan et al., 2018). In all cases, downregulating or stable mutants of these genes attenuated TSWV infection.

L439: Please move the arrows in Figure S1 so that the tip of each arrow points to the nucleocapsid aggregates.
Arrows have been moved towards the nucleocapsid aggregates.

L487: Please replace “factions” with “fractions”.
We corrected the typo. We apologize for the typo.

L507: Please replace “focussing” with “focusing”.
We appreciate spotting this typo and we have corrected it.

L536-540: Please provide evidence that the GO-enriched terms obtained in this section is related to virus assembly and cell-to-cell movement, or else discuss this in the Discussion.
We agree that we conclude too strongly that the protein are implicated in virus assembly and cell-to-cell-movement per se. We have changed the text to better indicate that the proteins are have a potential role in these processes based on their functional annotation (L537-542).

L541-563, L648-670: The comparison with previous studies of RBPome is lengthy and unclear in its purpose and should be eliminated.
We agree that the explanation of the used RBPomes are lengthy. We have in part rewritten these two sections without changing the content or its conclusions.

L560: What were the properties of the RNPs you analyzed? Viral particles, viral replication complexes, or something else? If they were viral particles, were the conditions of this experiment suitable for analyzing what you really wanted to analyze?
We kindly thank the reviewer 2 for these suggestions. In line with this comment, we have indeed confirmed that the RNPs fractions isolated from TSWV-infected N. benthamiana plants were infectious (with electron microscopy and via Koch’s postulates) in L438-447. Earlier others (Kikkert et al., 1997, Van Knippenberg et al., 2004) had already reported that this RNP  purification protocol yields functional RNPs. The EM revealed only RNPs  and we did not see any viral particles.

We fully agree with reviewer 2 that the study of a synchronised viral infection will be of high interest, but also difficult (as discussed in L973-983). As rub inoculation will not give a synchronized infection in all cells we expected that we will always isolate a mixture of recently infected cells and cells in a later stage of the infection due to viral spread from cell to cell. This would really require a transgenic approach with a molecular switch. Nevertheless, future research should investigate such dynamic changes in the TSWV host proteome. This manuscript paves this way by comparing the two different RNP systems.

L593-599: In this experiment, we cannot know if the function of the purified RNPs was intact or not. You need to show that the purified RNPs have replication activity in vitro, or the phrase needs to be changed.
We agree with the referee that we did not confirm the genomic replication of the viral S-genomic segment of RNPs purified from yeast as was shown by Ishibashi et al., 2017. We did not consider the replication properties to be required for our analyses, as we were foremost interested in host proteins that associate with the “synthetic” RNPs and not in polymerase activity per se. We did confirm the accumulation of the viral L protein, N protein and the genomic RNA (Figure 2B) and found that only in the presence of all three the protein levels of the N protein increased strongly in our RNP fraction. This is a strong indication for a stably formed macromolecular complex that purifies with the RNP fraction in density-based centrifugation steps. In addition, intact synthetic TSWV RNPs were observed via electron microscopy (Figure S1). These finding were described in L585-595.

L645-647: It is necessary to discuss the meaning that proteins associated with mRNA translation were co-purified with RNPs, even though the yeast replicon system supports only viral replication, but not transcription and translation of viral mRNA. It is also necessary to discuss the reason why candidate S genes were not obtained in this experiment.
The referee has a valid point that no S genes were obtained using the yeast RNPome approach. In L565-576, we justify our aim to identify conserved host factors via non-host yeast with the ultimate goal to translate these findings to a plant-host to identify potential S genes. As such, we first identified yeast proteins that co-purify with synthetic RNPs. Next, we determined functional homologs of these yeast proteins in N. benthamiana which we subsequently validated by gene silenced in N. benthamiana followed by a TSWV disease assay.

The allotetraploid genome of N. benthamiana consequently resulted in numerous gene copies. The observation that among the yeast-N. benthamiana selected candidate genes no S genes was identified does not imply that these are not required for TSWV pathogenicity. Nonetheless, we did identify four S genes from the viral RNPome purified from plants.

L812-813: Please replace “VIGS construct construct” with “VIGS construct”.
Table 1: Please change the commas to periods in log value and reduce the number of digits to three.
Corrected as requested.

Round 2

Reviewer 2 Report

The revised manuscript has been carefully rewritten and improved significantly. However, some of the points raised were not fully revised in the text even though they were mentioned in the author's response, and some of the evidence was still somewhat insufficient.
I recommend its acceptance after the minor points listed below are corrected.

L62-63: Due to insufficient description of the previous studies on the S genes for tospoviruses, please include in the text what is mentioned in the author's comments (shown below).
“The referred S genes are mutant versions of the 9-lipoxygenase (9LOX) and a-dioxygenases (a-DOX) in Arabidopsis (Garcia-Marcos et al., 2013), root hair defective 3 (rhd3) also in Arabidopsis (Feng et al., 2016), and suppressor of the G2 allele of skp1 (SGT1) in N. benthamiana (Quan et al., 2018). In all cases, downregulating or stable mutants of these genes attenuated TSWV infection.”

L566-572: Since there is little evidence that the GO-enriched terms obtained in this section is related to virus assembly or cell-to-cell movement, previous studies should be cited.

L934-942: Before discussing each of the genes identified in this study, it is better to describe the conclusion that four genes were identified as candidate S genes in the first paragraph of DISCUSSION.

L939-942: It is necessary to discuss the reason why candidate S genes were not obtained in the experiment using yeast. Please modify the content mentioned in the author's comments (shown below) as appropriate to fit the text and include it in the text. 
“No S genes were obtained using the yeast RNPome approach.”
“The allotetraploid genome of N. benthamiana consequently resulted in numerous gene copies. The observation that among the yeast-N. benthamiana selected candidate genes no S genes was identified does not imply that these are not required for TSWV pathogenicity.”
